# New Radiometric Approaches to Compute Underwater Irradiances: Potential Applications for High-Resolution and Citizen Science-Based Water Quality Monitoring Programs

**DOI:** 10.3390/s21165537

**Published:** 2021-08-17

**Authors:** Carlos Rodero, Estrella Olmedo, Raul Bardaji, Jaume Piera

**Affiliations:** 1EMBIMOS Research Group, Department of Physical and Technological Oceanography, Institute of Marine Sciences, CSIC, 37-49 Passeig Marítim de la Barceloneta, E-08003 Barcelona, Spain; jpiera@icm.csic.es; 2Department of Physical and Technological Oceanography, Institute of Marine Sciences, CSIC & Barcelona Expert Center, 37-49 Passeig Marítim de la Barceloneta, E-08003 Barcelona, Spain; olmedo@icm.csic.es; 3Marine Technology Unit, CSIC, 37-49 Passeig Marítim de la Barceloneta, E-08003 Barcelona, Spain; bardaji@utm.csic.es

**Keywords:** annular irradiance, water quality, marine citizen science, diffuse attenuation coefficient, oceanography, light

## Abstract

Measuring the diffuse attenuation coefficient ($K_{d}$) allows for monitoring the water body’s environmental status. This parameter is of particular interest in water quality monitoring programs because it quantifies the presence of light and the euphotic zone’s depth. Citizen scientists can meaningfully contribute by monitoring water quality, complementing traditional methods by reducing monitoring costs and significantly improving data coverage, empowering and supporting decision-making. However, the quality of the acquisition of in situ underwater irradiance measurements has some limitations, especially in areas where stratification phenomena occur in the first meters of depth. This vertical layering introduces a gradient of properties in the vertical direction, affecting the associated $K_{d}$. To detect and characterize these variations of $K_{d}$ in the water column, it needs a system of optical sensors, ideally placed in a range of a few cm, improving the low vertical accuracy. Despite that, the problem of self-shading on the instrumentation becomes critical. Here, we introduce a new concept that aims to improve the vertical accuracy of the irradiance measurements: the underwater annular irradiance ($E_{a}$). This new concept consists of measuring the irradiance in an annular-shaped distribution. We first compute the optimal annular angle that avoids self-shading and maximizes the light captured by the sensors. Second, we use different scenarios of water types, solar zenith angle, and cloud coverage to assess the robustness of the corresponding diffuse attenuation coefficient, $K_{a}$. Finally, we derive empirical functions for computing $K_{d}$ from $K_{a}$. This new concept opens the possibility to a new generation of optical sensors in an annular-shaped distribution which is expected to (a) increase the vertical resolution of the irradiance measurements and (b) be easy to deploy and maintain and thus to be more suitable for citizen scientists.

## 1. Introduction

The studies of light propagation and light field characteristics are crucial for understanding many physical and biological processes in the water bodies, driven by, or depending on, solar radiation [1], such as phytoplankton dynamics and surface bloom [2] or eutrophication [3]. This radiation at the sea surface is conventionally measured as spectrally resolved downward planar irradiance, $E_{d}\left(\lambda \right)$, and the attenuation of this quantity with depth (z) can be described by the diffuse attenuation coefficient $K_{d}\left(z,\lambda \right)$ [4]. This parameter is of particular interest in water quality monitoring programs because it represents a suitable proxy of water transparency [5] and it is related to light penetration and availability in aquatic systems [4,6]. Monitoring water transparency provides an indicator of the environmental status of the water body by providing information about phytoplankton concentrations or levels of dissolved organic and inorganic compounds. This is especially relevant in coastal areas and lakes that are strongly affected by human activities and rivers, winds, and waves.

Satellite-based ocean color sensors have been used to map optical properties of the ocean such as $K_{d}\left(z,\lambda \right)$ on local and global scales. Approximately 90% of the diffuse reflected light from a water body comes from a surface layer of water within a depth of 1/$K_{d}$ [7]. Therefore, $K_{d}$ is an important parameter for remote sensing reflectance of ocean colour satellite. With an increase in the remote sensing data availability over the past decade, there has been a rise in the in situ data available for calibration and validation of the satellite measurements [8]. However, the current satellite measurements for monitoring coastal and inland waters are still evolving and remain challenging because of the spatial scales that satellite measurements represent [9,10]. To improve data coverage in these zones, in situ irradiance measurements are still required. Furthermore, growing worldwide requires exploring cost-effective data acquisition to generate knowledge for sustainable natural resource management.

This need to develop novel approaches for monitoring environmental data is reflected in the recent growing attention toward citizen science [11]. Citizen science is an expanding practice in which scientists and citizens actively collaborate to produce new knowledge for science and society [12]. Volunteers participate in a wide range of projects to monitor aquatic ecosystems. For example, a project called Urbamar collects observational data on marine species in the Barcelona coastal area and creates a participative guide made with and by the people. Another project called Surfing for Science [13] assesses microplastic pollution in shoreline waters; the citizens participate in the project by collecting scientific samples while paddle surfing. Other citizen science projects involve monitoring environmental variables like water transparency and participants use optical instrumentation to do that. The Secchi Disk is a classical citizen science instrument used to measure the transparency of ocean and lake waters [14,15]. However, the quality of Secchi depth data varies from person to person as a function of vision, producing low-precision measurements. In addition, unlike electronic devices, this instrument lacks quality control of data and cannot set adjustment parameters. In recent years, electronic devices have appeared to monitoring water quality [16,17]. An example is the low-cost and DIY (Do-It-Yourself) moored system KdUINO, which allows measuring the diffuse attenuation coefficient parameter ($K_{d}$), related to water transparency [18]. The participation of citizen scientists in water quality monitoring can complement traditional monitoring methods and has other potential advantages such as lowering monitoring costs, significantly increasing data coverage, increasing social capital, enhancing support for decision-making, and enhancing the potential for knowledge co-creation [19].

There are some practical issues in acquiring in situ irradiance measurements, especially in areas where $K_{d}$ is not homogeneous. For example, in stratified water columns, the thin layers of phytoplankton can change across environmental gradients [20,21]. To detect and characterize these variations of $K_{d}$ [22] requires a system of optical sensors, ideally placed in a range of a few cm, improving the low vertical accuracy. However, there are inherent issues in the acquisition of in situ irradiance measurements mainly introduced by the influence of the instrumentation on in-water light fields: self-shading caused by the upwelling irradiance meter itself [23,24,25,26,27] and self-shading caused by a buoyed instrument [28,29]. Instrument self-shading can lead to an increase in the measurements’ uncertainty from a few percent to several tens of percent depending on the wavelength, the instrument radius, and the illumination conditions [30]. Moreover, there is an issue with instrument making. In the case of an instrument with multiple light sensors at different depths, these sensors must be attached to a structure that could also add a shadowing artifact to the integrated radiance value. This implies a limitation in the vertical distance of the sensors and thus a limitation of the accuracy of the measurements of $E_{d}$, especially in strongly stratified waters.

Here, we propose a new approach to integrate radiance: annular irradiance $E_{a}$ (see Figure 1). Therefore, instead of integrating the light arriving at the full upper semi-sphere, we propose integrating only the light that arrives at a specific ring of the light sphere. The choice of the annular angle in which the ring of sensors is set will determine the quantity of light arriving at the sensor and the shadow affecting the measurements. This new approach aims to (i) simplify the instrumentation design: instruments may be designed with a “tubular” shape. This makes them much easier to assemble, deploy, manipulate, and maintain. This simplicity makes $E_{a}$-based instruments ideal candidates to be used in citizen science-based water quality monitoring. (ii) Avoid self-shading: Sensors at a particular depth do not interfere with the rest of the sensors at different depths (this happens with $E_{d}$ based instruments). With this mechanism, the device works as a moored system, covering the spatial and temporal resolution of the water column. (iii) Provide very high vertical resolution measurements: As there are no potential self-shading effects, installing a large number of sensor units at different depths could be possible. This could be potentially useful for those observations that require concurrent high vertical resolution measurements.

The remainder of this paper is structured as follows. Section 2 presents the methods, numerical tools, and data sets used to develop and assess the radiance integration numerically in annular bands as annular irradiance ($E_{a}$) and its derived diffuse attenuation coefficient ($K_{a}$). Section 3 presents the results corresponding to the performance analysis of the $E_{a}$ and $K_{a}$ in terms of (i) the optimal integration annular angle and (ii) comparison and assessment of $K_{a}$ concerning the standard values of $K_{d}$. Section 4 provides a brief discussion on the results, by analyzing critical scenarios and finding possibles practical limitations and closes with the conclusions and outlooks.

## 2. Materials and Methods

### 2.1. Theoretical Basis

The fundamental measure of light energy in an aquatic system is the spectral radiance *L*, which in horizontally homogeneous water bodies is a function of time, depth, direction, and wavelength [4] that can be described by the following formula:(1)$$L\left(x,y,z,t,\theta ,\phi ,\lambda \right)=\frac{\Delta Q}{\Delta t\Delta A\Delta \Omega \Delta \lambda }$$
with ΔQ being the amount of radiant energy incident in a time interval Δt centered on time *t*, onto a surface of area ΔA located at position (x,y,z), and arriving through a set of directions contained in a solid angle ΔΩ on the direction (θ,ϕ) normal to the area ΔA, as produced by photons in a wavelength interval Δλ centered on wavelength λ [31]. The units of radiance are W m${}^{-2}$ nm${}^{-1}$ sr${}^{-1}$. One of the most commonly measured radiometric quantities is irradiance. The irradiance *E*, or flux density of radiant energy, is the integrated radiance on a unit area and is usually expressed as W m${}^{-2}$. The spectral downward plane irradiance $E_{d}$ is related to the spectral radiance, which measures photons traveling in all downward directions, but with each photon’s contribution weighted by the cosine of the photon’s incident angle θ [31].
(2)$$E_{d}\left(z,\lambda \right)=\int_{\phi =0}^{2\pi }\int_{\theta =0}^{\frac{\pi }{2}}L\left(z,\theta ,\phi ,\lambda \right)cos\theta |d\theta d\phi$$

In typical conditions, radiances and irradiances decrease approximately exponentially with depth. The downwelling diffuse attenuation coefficient $K_{d}$, in m${}^{-1}$, explains the concept of light extinction with depth of spectral downwelling plane irradiance $E_{d}$ [31].
(3)$$K_{d}\left(z;\lambda \right)=-\frac{1}{E_{d}\left(z;\lambda \right)}\frac{dE_{d}\left(z;\lambda \right)}{dz}$$

To know the vertical variation of $K_{d}$, $E_{d}$ needs to be measured within an infinitesimal range of depths. To overcome this obstacle, a common and useful practise is to calculate the diffuse attenuation coefficient between the irradiances measured over distant depths:(4)$${\bar{K}}_{d}\left(z_{1}↔z_{2};\lambda \right)=\frac{1}{z_{2}-z_{1}}\ln\left(\frac{E_{d}\left(z_{1};\lambda \right)}{E_{d}\left(z_{2};\lambda \right)}\right)$$
with $z_{1}$ and $z_{2}$ being different depths far apart to ensure reliable measurements of $E_{d}$ change. In addition, when there are vertical profiles of $E_{d}\left(z\right)$, $K_{d}\left(z1↔z2\right)$ is usually derived by linear regression analysis between $ln(E_{d}\left(z\right))$ and z [32], obtained as the negative of the slope of this linear regression. In this case, we assume that $K_{d}$ is a constant value through the depth range, and it is valued simply as ${\bar{K}}_{d}$. We will use ${\bar{K}}_{d}$ during the rest of this study.

### 2.2. Computational Fundamentals: HydoLight as a Numerical Tool

We use HydroLight (version 5.2) to define and assess the performance of the $E_{a}$. HydroLight is an example of a radiative transfer numerical model which computes radiance distributions and derived quantities given water column inherent optical properties and other oceanographic environmental conditions [33]. The HydroLight code employs mathematically sophisticated invariant embedding techniques to solve the radiative transfer equation and offers the possibility of performing numerical simulations in controlled environments. HydroLight performs a discretization of the two directions of working (θ(0≤θ≤180) and ϕ(0≤ϕ≤360), see Figure 2) with quad averaged radiances from a sphere. Therefore, this discretization allows the computation of the standard $E_{d}$ by the integration of the radiances of all the solid angles over the downward hemisphere, but also it allows to select of some specific quads and to integrate the corresponding radiances to derive a radiance over a configurable-by-the-user solid angle.

We will provide our results over the PAR (Photosynthetically Active Radiation), which is calculated by summing the contribution to the radiance of each of the bands which lie in the PAR range, that is, 400–700 nm. PAR is by definition a broadband quantity, expressed in mol quanta s${}^{-1}$ m${}^{-2}$.
(5)$$PAR\left(\vec{x}\;\right)=\int_{400\mathrm{nm}}^{700\mathrm{nm}}E_{d}\left(\vec{x}\;,\lambda \right)\frac{\lambda }{hc}d\lambda$$

In this case, PAR is calculated from $E_{d}$ irradiance. To convert $E_{d}$ in W m${}^{-2}$ in each band to mol quanta s${}^{-1}$ m${}^{-2}$, the per-band contributions are simply added up with unit conversion factors by 1/hc with *h* the Plank constant, *c* the speed of light in vacuum, and also Avagadro’s constant.

### 2.3. Definition and Computation of Annular Irradiance $E_{a}$ and $K_{a}$

We define the annular irradiance $E_{a}$ as the contribution of a selected θ band (by integrating all ϕ from $0^{\circ }$ to $360^{\circ }$) to irradiance onto a perpendicular sensor to the θ direction. We use HydroLight to integrate the annular irradiance by summing over the correct quads radiances and multiplying by the solid angles of them. In this case, the planar irradiance, which is the sum of radiance multiplied by solid angle and the cos(θ), will be the annular irradiance multiplied by a constant number.

We compute annular irradiance $E_{a}$ by using the following formula:(6)$$E_{a(\theta_{i})}=\sum_{j=1}^{24}L\left(\theta_{i},\phi_{j}\right)d\theta_{i}d\phi_{j}$$
where $L(\theta_{i},\phi_{j})$ is the radiance corresponding to the (i,j) quad and $d\theta_{i}d\phi_{j}$ factor is the solid angle of the i,j quad. HydroLight provides a discretization of the θ angle in bins of $10^{\circ }$ from $85^{\circ }$ to $5^{\circ }$, and two additional bins corresponding to the equator ($90^{\circ }$) and to the polar cap ($0^{\circ }$). Then,
(7)$$d\theta_{i}=cos\theta_{i}-cos\theta_{i-1}$$

The bins of ϕ are spacing at $15^{\circ }$ for all quads, so each ϕ bin has the same dϕ value:(8)$$d\phi \left(j\right)=\frac{(15deg)2\pi }{360}=0.2618$$

We use Equation (Equation 6) for computing annular irradiandes for $\theta_{i}=10,\ldots ,80^{\circ }$. We compute the corresponding diffuse attenuation coefficients, $K_{a(\theta_{i})}$ by using Equation (Equation 4) applied to the $E_{a(\theta_{i})}$ in the PAR, namely:(9)$${\bar{K}}_{a(\theta_{i})}\left(z_{1}↔z_{2};\lambda \right)=\frac{1}{z_{2}-z_{1}}\ln\left(\frac{E_{a(\theta_{i})}\left(z_{1};\lambda \right)}{E_{a(\theta_{i})}\left(z_{2};\lambda \right)}\right).$$

### 2.4. Data Sets Description

We use HydroLight to generate a total of 3024 different scenarios. We generate this wide variety of situations by changing:Water types: we consider in our study from extremely clear to extremely turbid water situations, simulating the kind of waters that could be encountered in estuaries and inland water bodies.Wavelengths: We consider wavelengths in between 400 and 700 nm at 5 nm intervals.Illumination conditions: we consider a solar zenith angle ranging from $0^{\circ }$ to $80^{\circ }$ in steps of $10^{\circ }$, and cloud coverage from 0% to 100% in steps of 20%.Depth resolution: we follow the depth resolution configurations described in Table 1.

Regarding the other parameters, we fix the wind speed to 0 m/s and assume that the water is infinitely deep homogeneous. The direct and diffuse solar irradiance were simulated using a semi-empirical sky model (the Radtran atmospheric irradiance model, developed by Gregg and Carder [34]), with the annual average sun-earth distance and ozone content of 300 DU as input. Raman scattering and chlorophyll and CDOM fluorescence were also included in all simulations. The input data concentrations of phytoplankton, colored dissolved organic matter (CDOM), and detritus/minerals were taken from the optical classification of lakes and coastal waters in Estonia and south Finland [35] (see Table 2).

From this configuration, the following data set were created:Data set 1: contains 3024 simulations, following the same parameters described above, with all the different water types (ultra clear, very clear, clear, moderate, turbid, very turbid, brown). Illumination conditions are considered as solar zenith angle range from $0^{\circ }$ to $80^{\circ }$ at $10^{\circ }$ intervals, and cloud coverage from 0% to 100% at 20% intervals. To optimize the number of simulations per water type and lighting conditions, it is considered two different values (maximum and minimum concentrations from Table 2) for each phytoplankton, CDOM, and minerals concentration.

From these data sets, we create different study cases depending on the analysis we want to address (see Table 3).

### 2.5. Methods to Analyze the Relations $E_{a}$-$E_{d}$, and $K_{a}$-$K_{d}$

A collection of methods is used to assess the correspondence between these variables, using the different scenarios described in Table 3. To assess the optimal $\theta_{i}$ that optimizes $E_{a(\theta_{i})}$ in terms of light acquisition and instrumental feasibility, we compute the ratio between $E_{a(\theta_{i})}$ and $E_{d}$ in logarithmic values at the following depths: 0.2, 0.5, 1.0, 1.5, 5.0, and 10.0 m. We represent the average ratio and the corresponding standard deviation at 1.5 m depth by using a bar plot. Once it is selected the optimal integration angle for annular irradiance ($\theta_{o}$), a scatter analysis is used to compare $K_{d}$ and $K_{a(\theta_{o})}$ to analyze the functional relationship between both. Then, we analyse how the diffuse attenuation coefficients change with different light conditions by modifying solar zenith angle and per cent of cloud coverage. The main goal is to analyze how the performance degrades as a function as these light conditions also degrade.

The results are plotted as heat-maps between the solar zenith angle and percent of clouds, all of them fixed at 1.5 m depth.

Regression analysis is used to estimate the strength and direction of the $K_{a}$ and $K_{d}$ relationship. We compute the correlation coefficients (*r*) (which measures the degree of association, with 0.05 probability level of significance) and slopes (*m*) (which measure the rate of change) of the regression line at 1.5 m. Both of them are obtained by computing a linear least-squares regression for two sets of measurements. Finally, the relative error, expressed in %, is calculated by the following equation:(10)$$\epsilon_{K}=\frac{{\tilde{K}}_{d}-K_{d}}{K_{d}}100$$
where ${\tilde{K}}_{d}$ is the estimated $K_{d}$ from $K_{a}$ values. We use the slope *m* resulting from the previous analysis to derive a ${\tilde{K}}_{d}$ from $K_{a}$:(11)$${\tilde{K}}_{d}=K_{a}/m$$
where *m* corresponds to the average of the slope of the regression line for each fixed depth.

## 3. Results

### 3.1. Optimal $E_{a}$ Integration Angle

We generate the annular irradiances from $E_{a10}$ to $E_{a80}$ from the data set described in the case study 1 (see Table 3).

As observed in Figure 3, $E_{a}/E_{d}$ in logarithmic values increases from $E_{a10}$ to $E_{a30}$ and then start to decrease at $E_{a40}$ by reaching very low values in the range of $E_{a60}$ to $E_{a80}$. This ratio has been tested at different depths (0.2, 0.5, 1.0, 1.5, 5.0, and 10.0 m depth), and the results are very similar. For this reason, we select the values only at one depth: 1.5 m.The standard deviation is widely distributed around the mean and indicates how wide it spreads out. Deep waters have a higher standard deviation in comparison with shallow waters.

To select the optimal annular angle, we need to take into account not only the optimization of $E_{a}$/$E_{d}$, but also that as closest the annular angle to the light sphere equator ($90^{\circ }$), the best performance in terms of avoiding self-shading effects and better the potential vertical resolution. The $E_{a}$/$E_{d}$ values at annular angles larger than $60^{\circ }$ are very low, indicating very low accuracy to the corresponding $E_{a}$ measurements. Therefore, we select $40^{\circ }$ as the optimal annular angle to measure $E_{a}$. For the rest of our study, the entire analysis is performed with $E_{a40}$ integration annular angle of 40 degrees. We also include in our analysis $E_{a50}$ because although it presents a decrease of accuracy with respect to $E_{a40}$, we want to assess a potential range of error in the placement of the sensors in the instrument.

### 3.2. Comparisons between $K_{d}$ and $K_{a40}$

Figure 4 represents a scatter plot between $K_{d}$ and $K_{a40}$ (left), and $K_{d}$ and $K_{a50}$ (right), both at 1.5 m (top) and 10 m (bottom) depths using data from case study 1 (see Table 3). In this case, $K_{a40}$ is strongly correlated with $K_{d}$ in all the water types, with a correlation coefficient value equal to 0.9828 at 1.5 m and 0.9919 at 10 m. The $K_{a50}$ is less correlated with $K_{d}$ than $K_{a40}$, especially in water types more turbid, although the correlation coefficient is also higher than 0.9 (0.9460 at 1.5 m and 0.9865 at 10 m).

### 3.3. Comparison between Different Lighting Scenarios

Figure 5 shows the correlation coefficient (left) and the regression line’s slope (right) between $K_{d}$ and $K_{a40}$ (upper) and $K_{d}$ and $K_{a50}$ (lower) as a function of the solar zenith angle and cloud coverage at 1.5 m depth. The correlation coefficient between $K_{d}$ and $K_{a40}$ remains constant, equal to 1, for almost all the different lighting scenarios. For solar zenith angles in the range of the same integration angle, $40^{\circ }$, the correlation coefficient decreases slightly, reaching values of 0.96. The correlation coefficient between $K_{d}$ and $K_{a50}$ follows the same structure as $K_{a40}$, with lower values in solar zenith angles between $40^{\circ }$ and $50^{\circ }$. In this case, all the correlation coefficient values are not as good as in the $K_{a40}$ correlation.

The regression line’s slope between $K_{d}$ and $K_{a40}$ presents variability in the range from solar zenith angles at $40^{\circ }$ and cloud coverage at 0–20%, with values equal to 0.81, to solar zenith angles at $0^{\circ }$ and cloud coverage at 0–20%, with values equal to 1.18. For the rest of the lighting scenarios the regression line’s slope remains stable in values close to 1.1. The regression line’s slope between $K_{d}$ and $K_{a50}$ presents variability in the range from solar zenith angles at $50^{\circ }$–$60^{\circ }$ and cloud coverage at 0–20%, with values equal to 0.79, to solar zenith angle at $80^{\circ }$ and cloud coverage at 40–80%, with values equal to 1.25. The rest of lighting scenarios the regression line’s slope remains stable in values close to 1.05–1.12.

Figure 6 shows the relative error between $K_{d}$ and $K_{a40}$ (left) and between $K_{d}$ and $K_{a50}$ (right) as a function of the solar zenith angle and cloud coverage at 1.5 m depth. By using the Equation (Equation 11) and the values of the linear regression from Table 4, we estimate the values of ${\tilde{K}}_{d}$ from $K_{a40}$ and $K_{a50}$, and we provide an estimation of the relative error by following Equation (Equation 10). In the case of the ${\tilde{K}}_{d}$ estimated from $K_{a40}$, for solar zenith angles from $0^{\circ }$ to $40^{\circ }$, and cloud coverage from 0% to 20%, it has a relative error in the range of 10% to 20%. In the other lighting scenarios, the relative error decreases to lower values until almost 0%. In the case of $K_{a50}$, the estimation of ${\tilde{K}}_{d}$ has relative errors higher than 20% at solar zenith angles from $0^{\circ }$ to $30^{\circ }$, and cloud coverage from 0% to 20%. In the rest of the lighting scenarios, the relative error is constant at 10% approximately. We have focused our analysis in 1.5 m depth, but as shown in Table 4 and Table 5 similar results are obtained when we consider different depths.

Finally, we analyze the relative error of ${\tilde{K}}_{d}$ as a function of the different types of water. Figure 7 shows that the estimated error of ${\tilde{K}}_{d}$ computed from $K_{a40}$ at 1.5 m depth has variations depending on which water type is analyzed. Clear waters have larger errors when the solar zenith angle is close to the zenith; however, turbid waters have errors focused on solar zenith angles close to the incident angle of $40^{\circ }$.

## 4. Discussion and Conclusions

This study presents the underwater annular irradiance $E_{a}$ as a new radiometric approach to compute underwater irradiances and its derived annular diffuse attenuation coefficient $K_{a}$ as an effective proxy to estimate the downwelling diffuse attenuation coefficient $K_{d}$ in the PAR region. We find that the optimal angle to measure underwater annular irradiance is at 40 degrees, which is the angle measured from the z-axis in spherical coordinates (see Figure 2). With this setup, light is captured by the sensor avoiding limitations due to instrument self-shading. The performances decrease with annular angles larger than 50 degrees, obtaining very poor performances with annular angles greater than 60 degrees. This degradation is probably caused by Snell’s law [36,37].

We observe a large correlation between $K_{d}$ and $K_{a40}$ in different water types (see Figure 4). In the case of the correlation between $K_{d}$ and $K_{a50}$, there are still correlations greater than 0.9. In this study, we analyze with special detail the performance at 1.5 m to simulate an instrument measuring the diffuse attenuation coefficient near the surface, as in real field conditions. We know that measures very close to the near-surface are affected by large light fluctuations caused by the surface waves [38]. Besides, the diffuse attenuation coefficient presents variability in response to changing solar altitude [39]. For this reason, in this correlation the solar zenith angle is set until $60^{\circ }$. The relationship between $K_{a40}$ and $K_{d}$ is robust when the light conditions change (see Figure 5). The larger differences of the correlation coefficient and the slope between $K_{a40}$ and $K_{d}$ occurs when the solar zenith angles directly affects the annular angle: the largest difference in the case of $K_{a40}$ happens when the solar zenith angle is incident at $40^{\circ }$, while the largest difference in the case of $K_{a50}$ happens when the solar zenith angle is incident at the range of 40${}^{\circ }$ to 50${}^{\circ }$.

Therefore, we derive empirical functions to estimate ${\tilde{K}}_{d}$ from $K_{a40}$ and $K_{a50}$ measurements, obtaining a relative error from this estimation and for each simulation. After that, we group the simulations in this case by different lighting scenarios. The estimation provides accurate measurements of $K_{d}$ (see Figure 6), which has relative errors below the 20% in the case of the estimations from $K_{a40}$ and below the 30% in the case of the estimations from $K_{a50}$. In both cases, the estimates of ${\tilde{K}}_{d}$ degrade for solar zenith angles close to the zenith. If we study this relative error grouping simulations by different water types, in the case of $K_{a40}$ (see Figure 7), we observe the largest errors in clear waters on solar zenith angles focused on zenith, and in turbid waters on solar zenith angles focused on incident angles at $40^{\circ }$.

As a result, the annular diffuse attenuation coefficient $K_{a40}$ allows the design of instruments expected to be particularly useful in those underwater environments where high vertical $E_{d}$ resolution is required. This design, ideally as a moored system tube, facilitates the instrumentation’s clean, critical when sampling different water columns and avoiding act as a disease vector [40]. Furthermore, devices based on this light-sensing approach are much easier to deploy and maintain. With these characteristics, the proposed work advances the current state-of-the-art of Marine Citizen Science as a DIY and low-cost sensor for water quality monitoring programs [41].

## Figures and Tables

**Figure 1 sensors-21-05537-f001:**
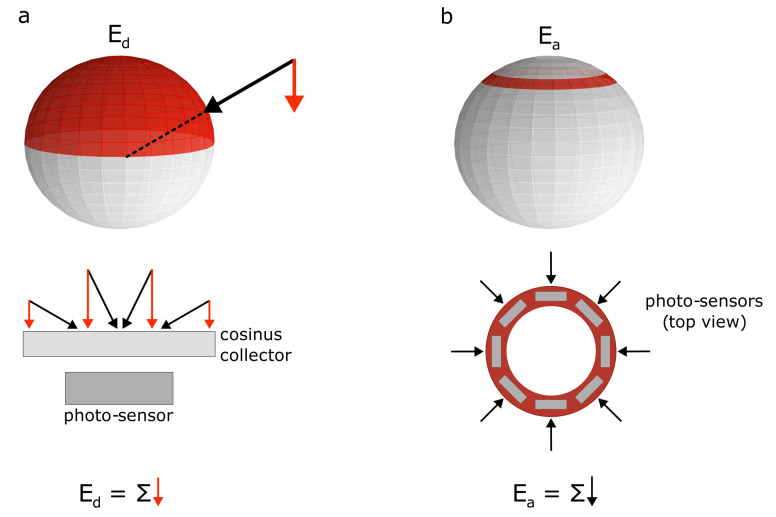
(**a**) Integrated radiances in the case of downward irradiance $E_{d}$. (**b**) Integrated radiances in the case of annular irradiance $E_{a}$.

**Figure 2 sensors-21-05537-f002:**
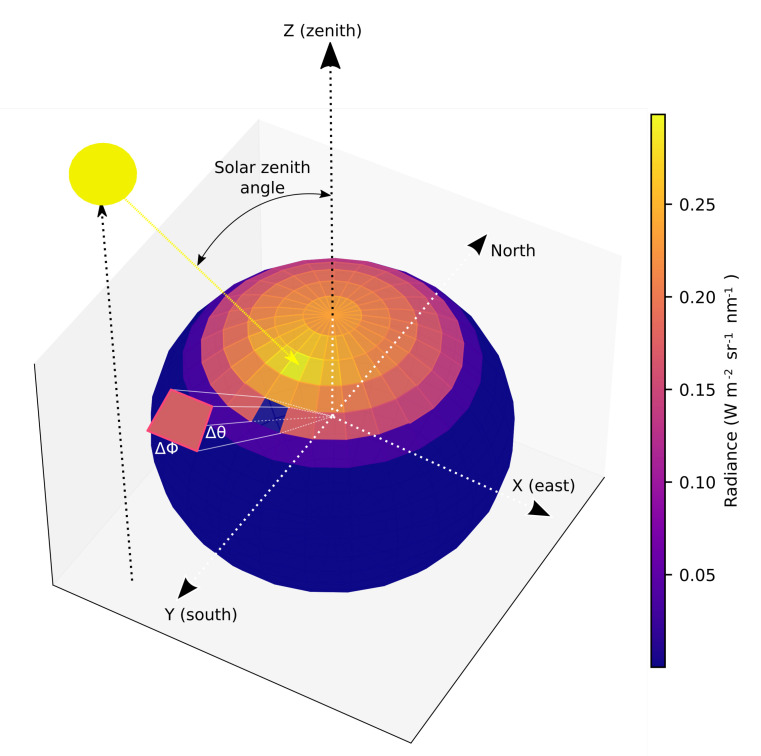
Radiance as a function of polar and azimuthal viewing directions, at 0.2 m depth and at wavelength 500 nm. Solar zenith angle is 30 degrees.

**Figure 3 sensors-21-05537-f003:**
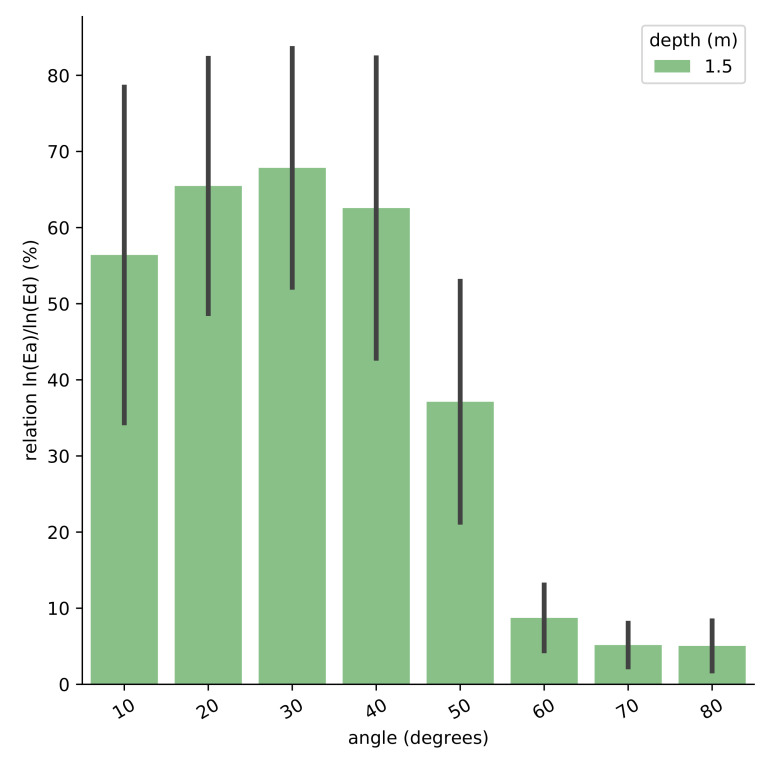
Comparison of the optimal $E_{a}$ integration annular angle in different water types, with solar zenith angle between $0^{\circ }$ and $60^{\circ }$ and cloud coverage between 0% and 100%.

**Figure 4 sensors-21-05537-f004:**
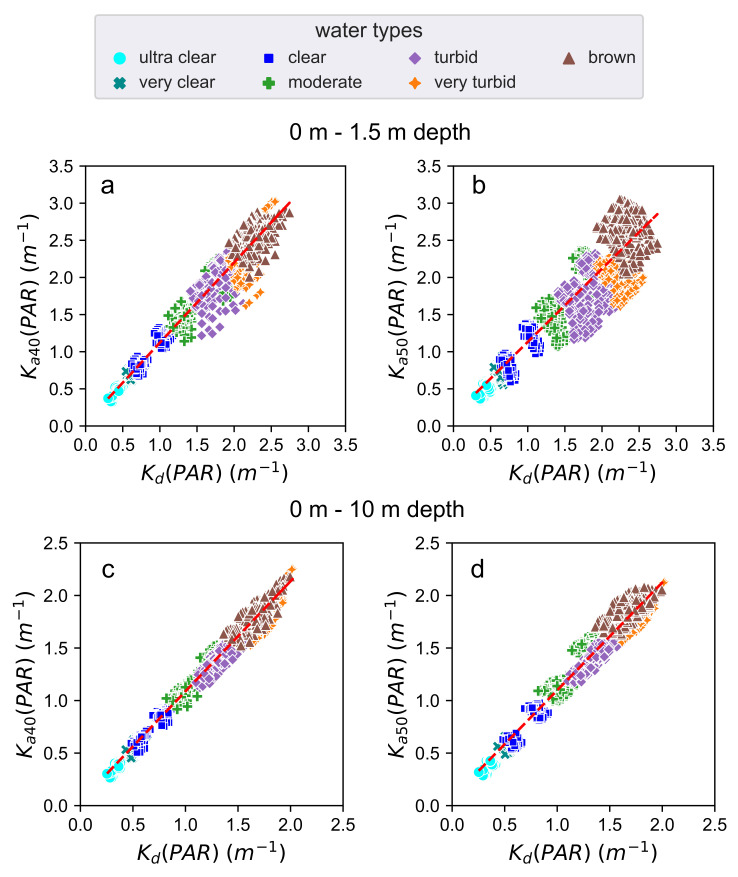
(**a**) Scatter plot between the $K_{d}$ and $K_{a40}$ computed in a range depth between 0 m and 1.5 m depth. (**b**) Scatter plot between the $K_{d}$ and $K_{a50}$ computed in a range depth between 0 m and 1.5 m depth. (**c**) Scatter plot between the $K_{d}$ and $K_{a40}$ computed in a range depth between 0 m and 10 m depth. (**d**) Scatter plot between the $K_{d}$ and $K_{a50}$ computed in a range depth between 0 m and at 10 m depth. All plots are configured in the PAR region, with solar zenith angle between $0^{\circ }$ and $60^{\circ }$, and cloud coverage between 0% and 100%.

**Figure 5 sensors-21-05537-f005:**
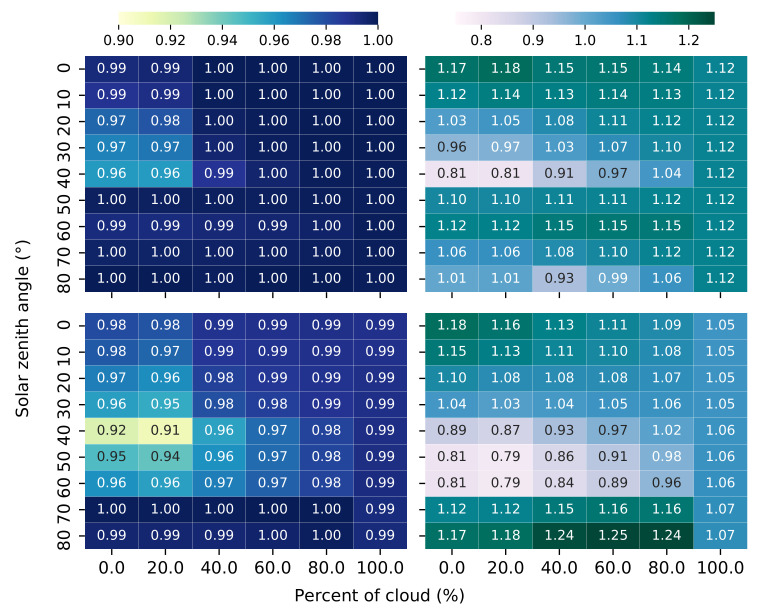
Correlation coefficient between $K_{d}$ and $K_{a40}$ (**upper left**). Regression line’s slope between $K_{d}$ and $K_{a40}$ (**upper right**). Correlation coefficient between $K_{d}$ and $K_{a50}$ (**lower left**). Regression line’s slope between $K_{d}$ and $K_{a50}$ (**lower right**). All the plots are configured modifying lighting scenarios as solar zenith angle and cloud coverage, at 1.5 m depth.

**Figure 6 sensors-21-05537-f006:**
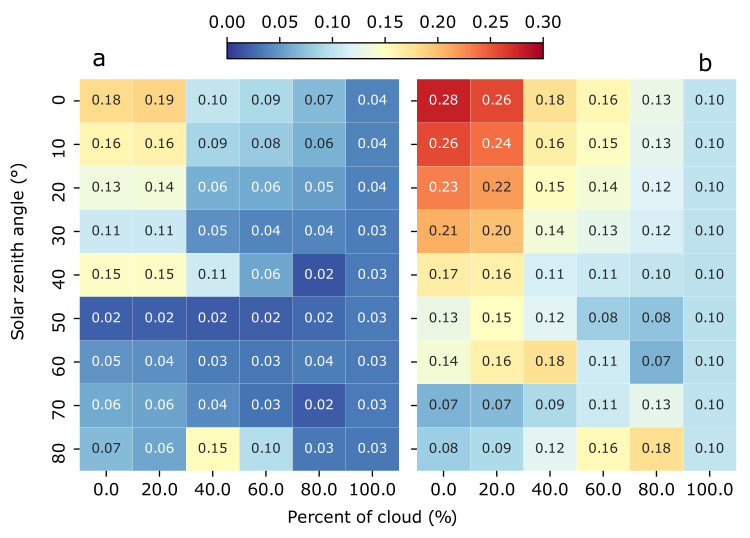
(**a**) Relative error between $K_{d}$ and $K_{a40}$. (**b**) Relative error between $K_{d}$ and $K_{a50}$. Both plots are configured modifying lighting scenarios as solar zenith angle and cloud coverage, at 1.5 m depth.

**Figure 7 sensors-21-05537-f007:**
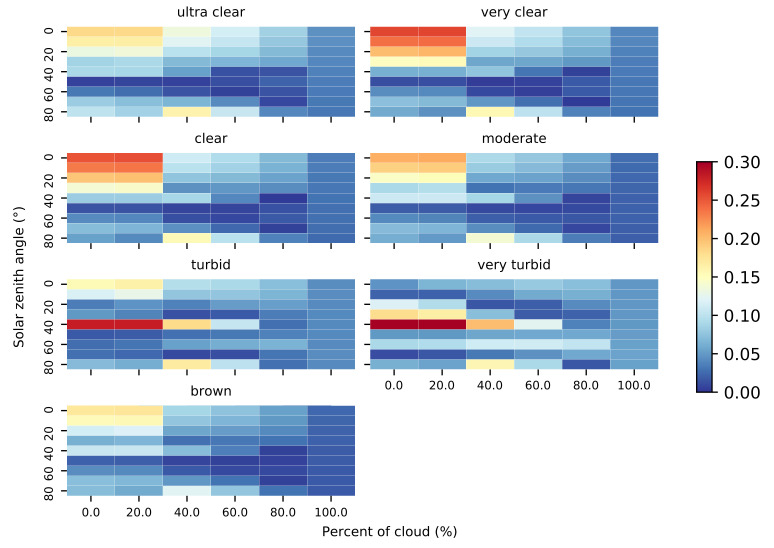
Relative error between $K_{d}$ and $K_{a40}$ depending on water types at 1.5 m depth, changing lighting scenarios as solar zenith angle and cloud coverage.

**Table 1 sensors-21-05537-t001:** Depth resolution configuration.

Value	Step
2 cm to 50 cm	2 cm
50 cm to 2 m	5 cm
2 m to 3 m	10 cm
3 m to 4 m	20 cm
4 m to 10 m	50 cm
10 m to 15 m	1 m
15 m to 20 m	5 m

**Table 2 sensors-21-05537-t002:** Characteristics of different type of water classes. Clear, moderate, turbid, very turbid, and brown classes are from the work in [35]. Very clear and ultra-clear classes are generated manually.

	chl (mg m${}^{-3}$)		cdom af(380)		mineral (g m${}^{-3}$)	
	Max	Min	Max	Min	Max	Min
ultra clear	0.0	1.0	0.0	0.6	0.0	0.8
very clear	1.0	3.0	0.5	1.5	0.5	1.5
clear	2.1	7.5	1.3	3.3	1.2	2.4
moderate	3.9	17.1	5.0	12.0	1.0	6.6
turbid	19.7	41.3	5.5	9.7	10.8	18.6
very turbid	65.2	67.6	6.1	6.7	30.3	38.7
brown	3.3	20.3	18.1	22.5	2.2	7.8

**Table 3 sensors-21-05537-t003:** Description of all the case studies.

	N. Simulations	Water Types	Solar Angle Range	Cloud Coverage	Figure-Table
1	2352	all	$\left[0^{\circ }:60^{\circ }\right]$	[0%:100%]	Figure 3 and Figure 4,
2	3024	all	$\left[0^{\circ }:80^{\circ }\right]$	[0%:100%]	Figure 5, Figure 6 and Figure 7, Table 4

**Table 4 sensors-21-05537-t004:** Average of slope for each depth range and for each diffuse attenuation coefficient used to calculate the relative error, from simulations generated in the case study 1.

	Depth-Range (m)					
	0–0.2	0–0.5	0–1.0	0–1.5	0–5.0	0–10.0
$K_{a40}$	1.125	1.095	1.084	1.079	1.070	1.057
$K_{a50}$	1.098	1.045	1.039	1.046	1.065	1.058

**Table 5 sensors-21-05537-t005:** Average of relative error for each depth range and for each diffuse attenuation coefficient, from simulations generated in the case study 1.

	Depth-Range (m)					
	0–0.2	0–0.5	0–1.0	0–1.5	0–5.0	0–10.0
$K_{a40}$	0.083	0.075	0.070	0.068	0.061	0.056
$K_{a50}$	0.140	0.154	0.149	0.138	0.112	0.095

## Data Availability

The work presented in this article is fully reproducible. The configuration files have been obtained with HidroLight (version 5.2) and the analysis of the data with Python (version 3.8.0). The following list shows the available links to the configurations, data and scripts used:The HydroLight configuration files: https://doi.org/10.5281/zenodo.5153536, accessed on 11 August 2021.The numerical results: https://doi.org/10.5281/zenodo.5041192, accessed on 11 August 2021.The code to generate, process and plot simulations: https://git.csic.es/36579996Z/pysimhydro/-/releases/v1.0.1, accessed on 11 August 2021. The HydroLight configuration files: https://doi.org/10.5281/zenodo.5153536, accessed on 11 August 2021. The numerical results: https://doi.org/10.5281/zenodo.5041192, accessed on 11 August 2021. The code to generate, process and plot simulations: https://git.csic.es/36579996Z/pysimhydro/-/releases/v1.0.1, accessed on 11 August 2021. The HydroLight configuration files and the numerical results are available under the terms
of the Creative Commons Attribution 4.0 International. The code to generate, process, and plot
simulations is licensed under the BSD-style license found in the LICENSE file in the root directory of
the source tree.

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
