# Peer review of "New Radiometric Approaches to Compute Underwater Irradiances: Potential Applications for High-Resolution and Citizen Science-Based Water Quality Monitoring Programs"

_sensors, 2021, doi:10.3390/s21165537_

Round 1

Reviewer 1 Report

The submitted manuscript proposed a new method to improve the vertical accuracy of the irradiance measurements. The work contributes a method from airborne LiDAR bathymetry data. The matter is of interesting! However,Figure 5,6,7 only presented water types at 1.5m depth, I would recommend to consider other depths range.

Reviewer 2 Report

Dear Authors,

In my opinion this is very interesting, scientifically sound manuscript. I read it with the great  interest. Also, the editorial quality is appropriate.

I decided accept the manuscript in the present state.

Best regards,

Reviewer

Reviewer 3 Report

This manuscript is very interesting that it gives a radiometric approach to measure the diffuse attenuation coefficient Kd. If the author could give the further data or presentation, it will be further improved.

  1. The HydroLight can be run by some initial data, which should be listed clearly. Although they are described in section of 2.4, it isn’t read clearly. If it is listed in a tabular form, it is better.
  2. In the section of “Water types”, the author should give the numerical descriptions of each type. And the following is to describe them in HydroLight.
